

# Osteogenesis imperfecta: potential therapeutic approaches

Maxime Rousseau[1,*], Jean-Marc Retrouvey[2,*] and Members of the
Brittle Bone Disease Consortium

[1] Faculty of Dentistry, McGill University, Montreal, QC, Canada
[2] Faculty of Dentistry, Department of Orthodontics, McGill University, Montreal, QC, Canada
* These authors contributed equally to this work.

## ABSTRACT

Osteogenesis imperfecta (OI) is a genetic disorder that is usually caused by disturbed production of collagen type I. Depending on its severity in the patient, this disorder may create difficulties and challenges for the dental practitioner. The goal of this article is to provide guidelines based on scientific evidence found in the current literature for practitioners who are or will be involved in the care of these patients. A prudent approach is recommended, as individuals affected by OI present with specific dentoalveolar problems that may prove very difficult to address. Recommended treatments for damaged/decayed teeth in the primary dentition are full-coverage restorations, including stainless steel crowns or zirconia crowns. Full-coverage restorations are also recommended in the permanent dentition. Intracoronal restorations should be avoided, as they promote structural tooth loss. Simple extractions can also be performed, but not immediately before or after intravenous bisphosphonate infusions. Clear aligners are a promising option for orthodontic treatment. In severe OI types, such as III or IV, orthognathic surgery is discouraged, despite the significant skeletal dysplasia present. Given the great variations in the severity of OI and the limited quantity of information available, the best treatment option relies heavily on the practitioner's preliminary examination and judgment. A multidisciplinary team approach is encouraged and favored in more severe cases, in order to optimize diagnosis and treatment.

Corresponding author
Maxime Rousseau,
maxime.rousseau2@mail.mcgill.ca

## INTRODUCTION

Osteogenesis imperfecta (OI), or "brittle bone disease," is caused by mutations in the collagen type I genes COL1A1 and COL1A2 (or other collagen genes for rarer types of OI), causing production of a defective collagen type I that results in significant alterations in different tissues in the body. However, the defective collagen affects bones more than it does most other organs (*Forlino & Marini, 2016*). Typical signs of OI are small stature, fragile bones, scoliosis, blue sclera and Shields type I dentinogenesis imperfecta (DI). Four main types of OI have been described by *Sillence, Senn & Danks (1979)*, and this classification is still the most commonly used today, though more clinically defined types were later described by *Rauch & Glorieux (2004)* (Table 1).

**Table 1 Characteristics of subjects affected by OI according to types.**

| | Characteristics |
|---|---|
| Type I: Mild non-deforming Osteogenesis Imperfecta | Normal height, or mild short stature; blue sclera; low rate of dentinogenesis imperfecta (20%) |
| Type II: Perinatal lethal | Multiple rib and long bone fractures at birth; pronounced deformities; broad long bones; low density of skull bones on radiograph, dark sclera |
| Type III: Severely deforming | Very short; triangular face; low ears, severe scoliosis; greyish sclera; dentinogenesis imperfecta (80%) |
| Type IV: Moderately deforming | Moderately short; mild to moderate scoliosis; greyish or white sclera; dentinogenesis imperfecta (60%) |
| Type V: Moderately deforming | Similar fracture incidence and long-bone deformities to type IV; Hyperplastic callus formation; calcification of the interosseous forearm membrane |
| Type VI: Moderately to severely deforming | Moderately short; scoliosis; accumulation of osteoid in bone tissue, fish scale pattern of bone lamellation; white sclera; no dentinogenesis imperfecta |
| Type VII: Moderately deforming | Mild short stature; short humeri and femora; white sclera; no dentinogenesis imperfecta |

It is important to note that these OI types are not named in terms of severity. Rather, the severity of OI increases as follows: type I, type IV, type III, and type II. OI types V, VI, and VII are quite rare, and their disease severity tends to be similar to OI type IV. It is important to realize that disease severity is a continuum, and that classification into types is not an exact science. Even within a particular type of OI, there is great variability in disease characteristics. Thus, the degree of severity of manifestations such as DI, scoliosis, and bone fragility can be very different from one subject to another, even if both subject present the same OI type.

## SURVEY METHOD

Literature searches were performed with "dental" and "osteogenesis imperfecta," as in dental manifestations of OI. The queries were performed through the Biosis (78 articles), Embase (362 articles), Medline (245 articles), and Web of Science (189 articles) databases. Altogether, the search yielded 874 articles. Relevant articles were screened by examination of the titles and abstracts. Case studies, case reports, or animal studies were not included in the final selection. A total of 80 articles were retained for the literature review and read in their entirety.

## MEDICAL MANAGEMENT

There is currently no cure for OI, as the underlying genetic cause cannot be directly addressed. However, symptomatic treatment options are available and are used according to clinical disease severity. Subjects with mild OI type I may not need any medical treatment if they are fully mobile and experience few fractures. However, many of the more severely affected subjects receive intravenous (IV) bisphosphonates to increase bone mineral density and to reduce fracture rates (*Glorieux, 2008*). In growing children, this treatment also helps to reshape vertebrae which were compressed by fractures.

Bisphosphonate therapy seems to be less effective in adults than in children (*Shapiro, Kantipuly & Rowe, 2010*).

Apart from drug therapy, subjects with decreased mobility due to leg deformities often require surgical interventions to straighten out long bones with metal rods that are placed into the bone marrow cavity (*Ruck et al., 2011*). Subjects with scoliosis may need spinal fusion surgery. Physiotherapy programs and regular physical exercise are other key components of subject management (*Rauch et al., 2002*; *Cheung & Glorieux, 2008*).

## DENTAL MANIFESTATIONS AND TREATMENTS

### Dentinogenesis imperfecta and its management

Osteogenesis imperfecta subjects present more dental problems than the average population (*Saeves et al., 2009*). The main dental manifestation of OI is DI (*Barron et al., 2008*; *Levin et al., 1983*; *Vital et al., 2012*), but it is not visibly present in every subject affected by OI. The prevalence of DI varies by Type, from 21% to 73%, as reported in the literature (*Majorana et al., 2010*). DI is present in 25% of the OI type I population (*Paterson, Mcallion & Miller, 1983*), 60% of OI type IV, and up to 80% of OI type III. OI type V and VI do not seem to be affected by DI (*Schwartz & Tsipouras, 1984*). The DI present in OI subjects is classified as Shields DI type I (*Shields, 1983*) and is characterized by yellow to bluish-brown discoloration of teeth due to abnormal dentine (*Jindal et al., 2009*; *Shields, Bixler & El-Kafrawy, 1973*) and short roots. Initially, the primary pulp chambers are uncharacteristically large, but they will calcify fairly rapidly. The enamel, although normal, is prone to fracture due to the deficient dentinoenamel junction, which is smooth instead of scalloped (*Delgado et al., 2008*).

Studies suggest that teeth affected by DI are not at greater risk of developing carious lesions (*Sapir & Shapira, 2001*). This slow progression of caries is thought to be caused by the random nature of the dentinal tubules and the fact that there are fewer tubules (*Teixeira et al., 2008*). The primary dentition is usually more vulnerable to breakdown by DI than the permanent, although permanent teeth are still prone to deterioration over time (*Waltimo, Ojanotko-Harri & Lukinmaa, 1996*). Although DI is not always clinically detectable in all OI subjects, some characteristics may still be observed due to the large degree of variation in the severity of the disease (*Levin et al., 1983*; *Waltimo, Ojanotko-Harri & Lukinmaa, 1996*). Precautions regarding restorative dentistry should be taken in all OI subjects, as their teeth may still be affected despite a negative clinical diagnosis of DI (*Lund et al., 1997*). Obliteration of the pulp chambers, short roots, and bulbous crowns may compromise the restorability of the dentition.

For subjects affected with moderate to severe OI, and especially if combined with DI, a multidisciplinary team approach is the current recommendation, to ensure that the diagnosis is accurate and that treatment approaches are optimized (*Cheung et al., 2011*). The dental team should include a pediatric dentist, a periodontist, an oral surgeon, an orthodontist, and a restorative dentist. The diagnostic procedure involves a thorough clinical examination, panoramic and periapical radiographs, and a complete family history.

Before contemplating restorative interventions, a thorough radiological and clinical examination of the dentition must be performed to assess the degree of pulpal calcification, the shape of the teeth, and the length of the root. It is also important to consider the clinical restorability of each tooth with a prognosis for long-term success, as well as to maintain adequate vertical dimension of occlusion (VDO) until natural exfoliation occurs. Also, it is important to consider the subject's need for serial magnetic resonance imaging due to the medical condition. Restorative options that are favorable for this type of imaging may be indicated (*Sapir & Shapira, 2001*).

Histologic analysis of the primary dentition upon exfoliation is indicated, if possible, when the subject is treated in a hospital environment. This procedure will allow for an assessment of the severity of the DI and help the practitioner better evaluate the subject's condition (*Wright & Gantt, 1985*).

Benefits of early treatment in the primary dentition of children with DI (*Herold, 1972*) include:

- Maintaining dental health while preserving tooth structure
- Providing ideal aesthetic appearance
- Preserving function of dentition and maintaining the VDO
- Avoiding premature loss of primary teeth while maintaining the stability of arch length
- Establishing a dental home which allows for building a relationship with a dental team and reinforces the importance of oral health

At present, the consensus is that, if needed, treatment should be initiated at an early stage, as the subject's dentition tends to deteriorate with an associated breakdown of tooth structure over time (*O'Connell & Marini, 1999*). Typically, treatment should be completed before or at the time of the full primary dentition eruption. Most often, general anesthesia is necessary due to the inability to use other advanced behavior guidance techniques in children with OI. Special care is needed during anesthesia and dental treatment due to the subject's need for immobilization and risk of fracture. Depending on the severity of DI presentation, a one- or two-stage approach can be used. Ideal treatment would consist of a one-stage approach and treating all primary teeth once they have erupted and are in occlusion. However, if the condition needs treatment prior to all teeth being erupted, a two-stage approach can be used (*Herold, 1972*). A stainless steel crown or a full ceramic crown should be placed if the tooth shows signs of weakness and if it starts to fracture. The best option will usually be the one that puts the least amount of stress on the tooth and supporting bone (*Malmgren & Lindskog, 2003*; *Teixeira et al., 2008*). Full coverage restorations are the treatment of choice, since intracoronal restorations are very poorly retained in DI teeth and may actually promote fractures (*Majorana et al., 2010*). Clinically, all primary teeth show signs of breakdown from DI, with primary canines being least affected. Generally, the permanent dentition is less affected (*O'Connell & Marini, 1999*). Dental sealants placed on newly erupted permanent molars may offer supplemental protection and are strongly recommended (*Delgado et al., 2008*; *Leal et al., 2010*; *O'Connell & Marini, 1999*).

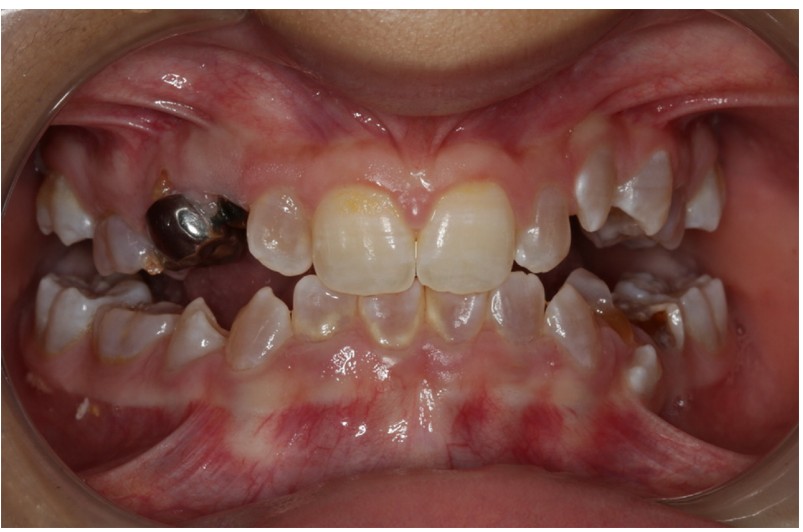

**Figure 1 Subject affected by OI type IV with DI.** This picture was selected from the authors' archives.

In young adults that have a permanent dentition affected by moderate to severe DI, full coronal coverage is preferred when clinical or radiological signs of post-eruptive breakdowns are diagnosed (*Siadat, Alikhasi & Mirfazaelian, 2007*). This treatment method may enhance survival/retention.

Severely fractured or non-restorable teeth are better off extracted, as tooth prognosis becomes guarded/poor, thus limiting the restorative treatment options. In some cases, if the subject is asymptomatic, or if the medical condition does not allow dental treatment and no clinical or radiographic signs of infection exist, it can be indicated to monitor the dental condition. Teeth deemed non-restorable or those not candidates for pulpal therapy should be extracted, with consideration given to space maintenance. Teeth that are candidates for pulpal therapy might fulfill the requirements for full-coverage restorations and frequent radiographic follow-up. Endodontic treatment carried out on teeth of subjects affected by DI may prove difficult given the altered morphology and calcification.

Osteogenesis imperfecta subjects not diagnosed with DI may still present with abnormal dental development that may weaken their structure. Short root and slender shape, abnormal pulpal calcification, or other morphological anomalies are significant clues that the dentition is not normally developed (*Leal et al., 2010*) (Fig. 1). The treatment plan for such individuals should depend on comprehensive clinical observations by the dentist and should be correlated with the severity of the OI type.

## Delayed ectopic eruption patterns and management

One of the issues encountered when dealing with young subjects affected by OI is delayed and ectopic eruptions (*Malmgren & Norgren, 2002*). These manifestations may be accentuated by bisphosphonate treatments, as the physiologic mechanism of bone resorption around the dental sac is disturbed either by the OI condition or by the use of bisphosphonates (*Kamoun-Goldrat, Ginisty & Le Merrer, 2008*; *Malmgren et al., 2016*).

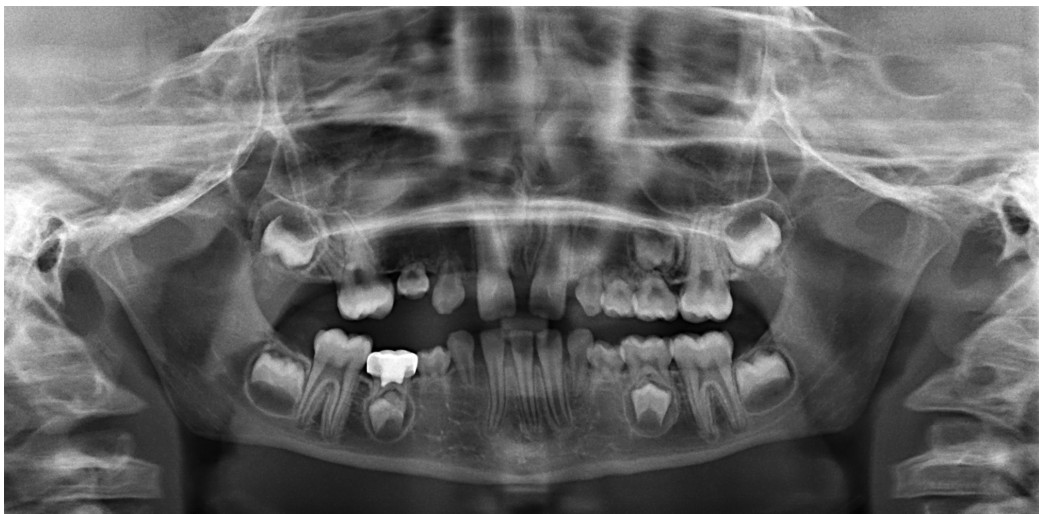

**Figure 2 Panoramic radiograph of a subject affected by OI type III with missing teeth and DI.** This picture was selected from the authors' archives.

Missing and impacted teeth are also common findings, especially in type III OI associated with DI (*Malmgren et al., 2016*) (Fig. 2).

The best approach in these cases will depend strongly on the severity of the subject's condition and the position of the impacted tooth or teeth and missing teeth. OI type I subjects presenting with impactions may be treated with conventional orthodontics, while more severe types of OI which present complex impactions may require surgical assistance (*Chang, Lin & Hsu, 2007*). It is safe to try to limit surgery, as their success rate and healing process may be affected by the OI severity and bisphosphonate treatment (*Schwartz & Tsipouras, 1984*).

Osteogenesis imperfecta type I cases present with the least challenges, while OI type III affected by DI tends to create much more difficult clinical situations (*Rizkallah et al., 2013*). Impacted teeth, especially upper canines, may be brought into the arch during orthodontic treatment when performed on OI type I subjects. It may be advisable to postpone the removal or surgical exposure of asymptomatic, impacted teeth in severe OI subjects, to avoid potential surgical complications and further loss of alveolar bone (*Malmgren & Norgren, 2002*).

## Malocclusions management in OI

Subjects affected by OI will show craniofacial manifestations of the disease to varying extents, depending on the severity of their condition (*Rizkallah et al., 2013*; *Vital et al., 2012*). Triangular-like face shape and low ears are the most widely reported facial bone features found in OI subjects, especially in types III and IV (*Rauch & Glorieux, 2004*). This phenomenon was investigated on an animal model and partially explained in a study conducted by *Eimar et al. (2016)*. The characteristics were attributed to the disproportionate relationship between the subject's small chins and large malar bones. These subjects very often show signs of restrained vertical dimension, flattened cranial base, relative prognathism, and more forward, counterclockwise mandibular growth

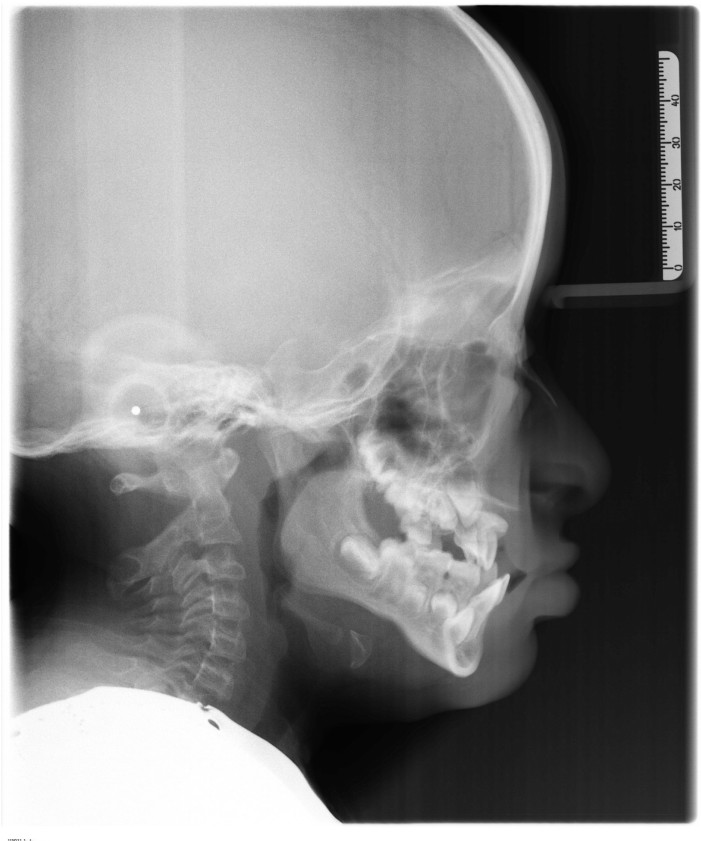

**Figure 3 Young subject affected by OI type III with a severe class III malocclusion.** This picture was selected from the authors' archives.               

(*Chang, Lin & Hsu, 2007*). Studies (*Chang, Lin & Hsu, 2007*; *Rizkallah et al., 2013*) have shown that OI subjects tend to be prone to more severe and unique malocclusions than unaffected subjects. It was shown that Class III malocclusions occur more readily in subjects affected by OI (*Schwartz & Tsipouras, 1984*; *Waltimo-Siren et al., 2005*) (Fig. 3). Lateral open bites, a rare finding in the general population, as well as anterior crossbites, have a much higher incidence in the OI population than in unaffected subjects (*Rizkallah et al., 2013*). However, few reports of successful orthodontic treatments are present in the literature (*Hartsfield, Hohlt & Roberts, 2006*). OI type I subjects who present moderate malocclusions and no severe DI may be treated cautiously, especially if orthognathic surgery is contemplated (*Tashima et al., 2011*). Not enough clinical data is available about outcomes of orthodontic treatments in moderate to severe OI subjects to issue clear treatment guidelines. Intervention at an early age may be best, since the subject's dentition and malocclusion tend to worsen over time. It is also believed that the best option is usually the one that puts the least amount of stress on teeth and bone, as they are more fragile (*Malmgren & Lindskog, 2003*; *Teixeira et al., 2008*). One approach used in non-OI subjects presenting with anterior and posterior crossbites consists of rapid maxillary expansion and a protraction face mask (*Kapust, Sinclair & Turley, 1998*; *Kim et al., 1999*). This procedure has been time-tested and is efficient in the unaffected

population (*Mandall et al., 2016*). However, this procedure may be cause for concern in OI subjects, as the development of the bone mass in the maxilla is defective in all OI subjects, particularly type III and IV. The application of orthopedic forces to fragile bones and teeth with compromised sutural growth potential may be detrimental to the treatment outcome. The transverse discrepancy found in severe OI subjects may be too large to benefit from rapid maxillary expansion significantly, and root morphology may not respond well to the large forces applied. In conclusion, maxillary expansion in OI subjects must be approached with caution, especially in moderate to severe OI. It is probably contraindicated in severe OI–DI subjects. The vertical dimension, in the case of lateral open bites, should be prioritized to restore occlusal contacts in the posterior segments, even if the dentition is to remain in crossbite after treatment. The anteroposterior correction may be attempted with light intermaxillary elastics, which put much less stress on the dentition and bony structures than a protraction device. Orthognathic surgery, if it involves a maxillary repositioning, is not routinely recommended due to high morbidity, potential excessive bleeding, and relapse (*Hartsfield, Hohlt & Roberts, 2006*; *Lewis & Stoker, 1987*). The goal of orthodontic treatment in these subjects is not to correct minor aesthetic imperfections, but to restore a functional occlusion. It is also important to select subjects carefully. Appliances that put heavy stress on bones and teeth, such as a face mask, are strongly discouraged. Clear aligners, as tested by the author, may prove to be the best alternative in applicable cases (Fig. 4).

## Osteonecrosis of the jaw—potential risks

There were initial concerns regarding bisphosphonate treatments, especially IV infusions administered to subjects affected by OI, related to the possibility of subjects developing osteonecrosis of the jaw following dental procedures or even simple extractions (*Kos et al., 2010*; *Ruggiero et al., 2004*). Bisphosphonates decrease osteoclasts' activity, which in turn affect the mechanism of bone repair, maintenance, and remodeling. High dosage of bisphosphonates can also reduce blood supply to a wound, creating a potential risk for Osteonecrosis of the jaw (ONJ) (*Schwartz et al., 2008*).

Four studies looked into the risks of osteonecrosis of the jaw in subjects affected with OI. One of these studies followed five subjects who had at least one tooth extraction, and none showed evidence of ONJ four or sixty months after extraction (*Johnson & Hodgson, 2010*). Another study followed fifteen subjects who had extractions, mostly of primary teeth, during or after completion of the bisphosphonate treatment. No complications were reported after the extractions, and the healing times were within the expected norm (*Schwartz et al., 2008*). The third study is a systematic review that looked at five publications. This review covered a total of 163 subjects who were subjected to dental surgery (surgical and nonsurgical dental extractions and/or manipulations of bone) while undergoing bisphosphonates treatment. No osteonecrosis of the jaw was reported (*Chahine et al., 2008*; *Hennedige et al., 2013*; *Maines et al., 2012*). Even if ONJ has not been linked to bisphosphonate-treated OI subjects, it is recommended that extractions not be performed immediately prior to or immediately after bisphosphonate administration. Delaying infusion of IV bisphosphonate has been found to cause increased bone pain

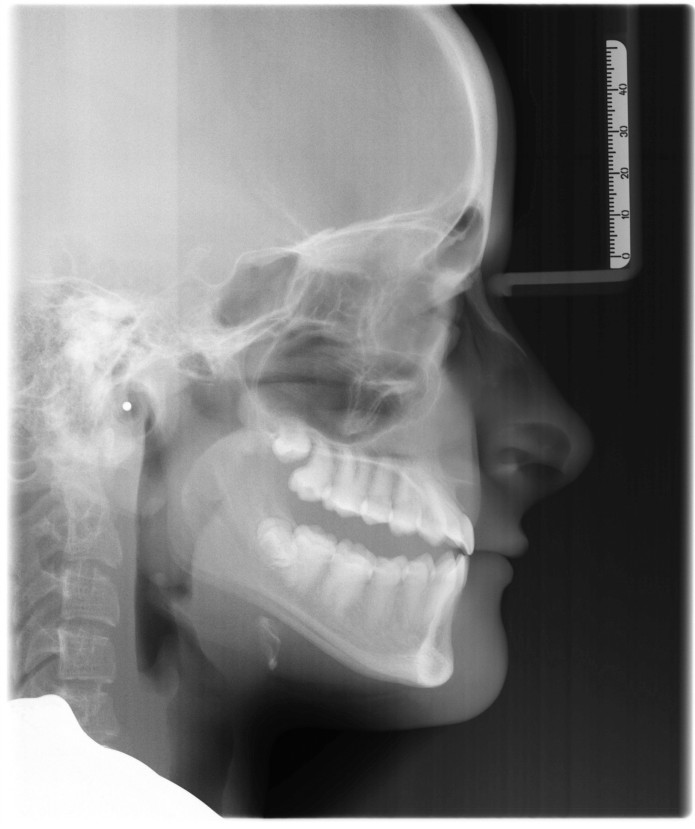

**Figure 4 Lateral open bite on a subject affected by OI type IV treated with the Invisalign appliance.** This picture was selected from the authors' archives.

for some subjects. If treatment is stopped for long enough, there may be a higher risk of having a fracture (*Rauch et al., 2007*). As the half-life of bisphosphonates is very long (*Fleisch, 1997*), waiting for more than a week before dental procedures will not significantly affect the outcome or lower the morbidity of the procedure. We recommend 24–48 h to ensure that the bisphosphonates are absorbed and out of the bloodstream. It is important to note that withdrawing the medication is not thought to make any difference in the surgical outcome. Bisphosphonates have a 10-year half-life and will accumulate in the bone for up to three half-lives, or 30 years (*Fleisch, 1991*).

Osteonecrosis of the jaw usually occurs when high dosages of IV bisphosphonates are administered. However, information is still limited regarding OI subjects, and we have yet to discover the risks regarding dental implants in long-term BP treatment (*Schwartz et al., 2008*).

## Dental implants

Dental implants are a popular modality to replace missing teeth, and they have become very reliably used in the general population. OI type I may receive dental implants if their overall periodontal condition is not compromised and bone density at the site is adequate to accommodate the implant. Cone beam computerized tomographic
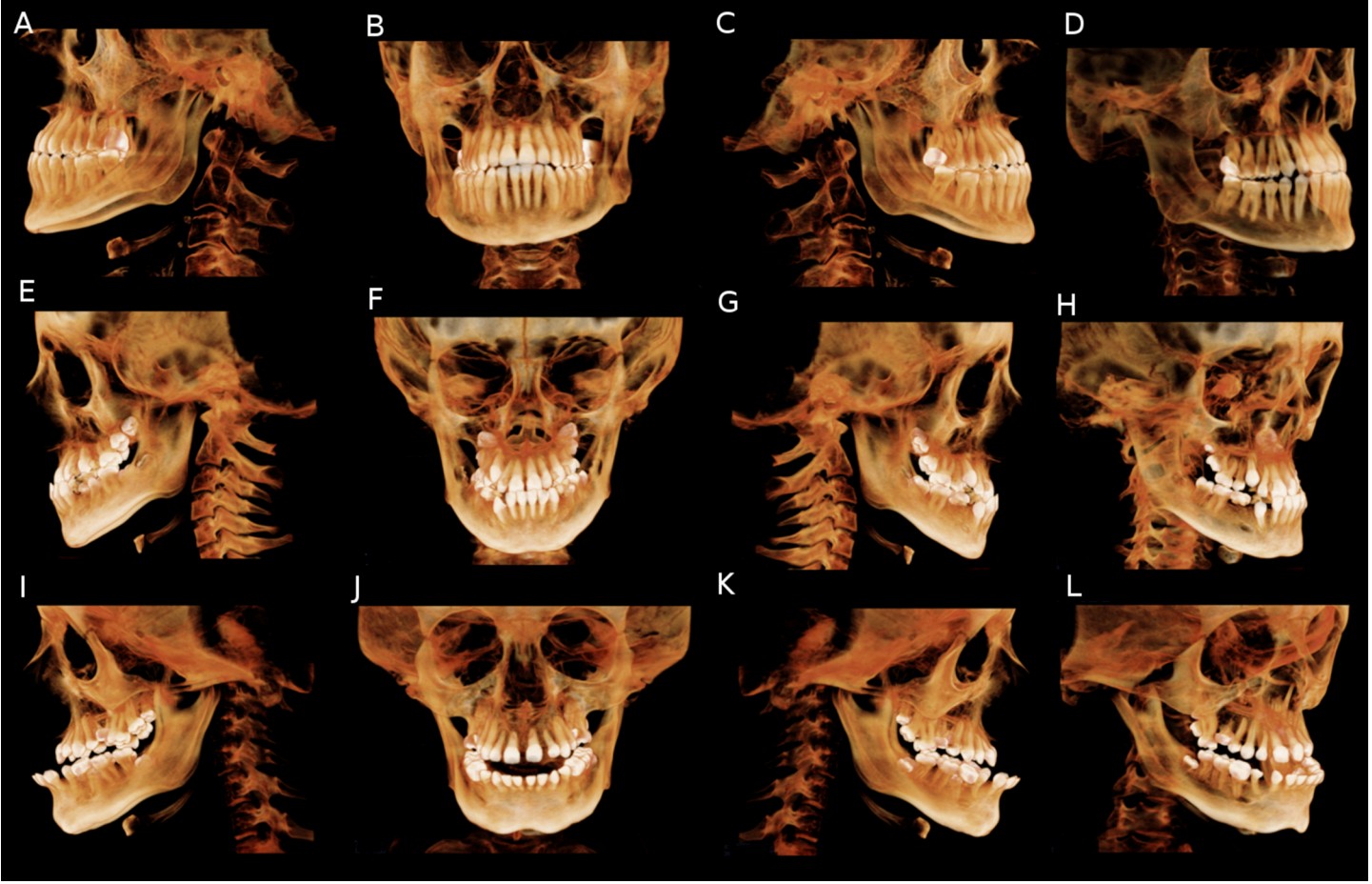

**Figure 5  CBCT volume rendering of subjects affected by OI to different degrees of severity.** (A–D) OI type I subject presenting with end to end anterior relation, full cusp class III malocclusion and without sever skeletal deformities. (E–H) OI type type IV subject presenting with missing teeth, class III malocclusion with negative overjet, asymmetry and slight skeletal deformities. (I–L) OI type III subject presenting with severe class III malocclusion, anterior open bite and pronounced skeletal abnormalities. This picture was selected from the authors' archives.

(CBCT) radiography in addition to a thorough clinical examination may assist the clinician in assessing the amount and quality of bone present in the edentulous area before planning the insertion of dental implants in this area. More severe cases such as type III and IV OI subjects present more risks, as bone density and healing potential may be compromised. A few cases have been reported in the literature showing success in integration of implants. A publication described a subject suffering from OI type IV with complete upper and lower dentures supported by six maxillary posterior and five mandibular anterior osseointegrated implants (*Prabhu et al., 2007*). There was also a report of successful implant therapy in a subject affected by type III OI where four implant supported metal ceramic crowns were placed as a treatment for mandibular anterior teeth agenesis (*Wannfors, Johansson & Donath, 2009*).

Cone beam computerized tomographic studies should be considered as part of the preliminary investigations for implant therapy. The main advantages of this type of

imaging is that it enables practitioners to visualize skeletal asymmetry in 3D space as well as teeth and root orientation (Fig. 5).

Moreover, CBCT have the ability to allow for accurate measurement and quantification of the craniofacial anomalies present in OI subjects. Even if OI subjects present with specific craniofacial anomalies such as a retrognathic and hypoplastic maxilla, a prognathic mandible and asymmetric cranial features, preliminary studies have failed to directly craniofacial malformations with OI types. More studies at the genetic levels and involving potential correlation between observed and measured craniofacial deformities and the genetic "mapping" of OI subjects should be performed.

To our knowledge, no prospective or retrospective study of a large enough sample has been published. Our recommendation would be to approach implant dentistry with caution in cases of moderate to severe OI, as the failure rate may be higher than in the unaffected population due to the inherently poor bone architecture and remodeling capacities (*Hartsfield, Hohlt & Roberts, 2006*). The risk of failure and complications may be increased with prolonged use of IV bisphosphonates.

## CONCLUSION

Given a great deal of variability in the degree of severity of OI subjects, treatment plans may vary. It is the practitioner's responsibility to ensure that the treatment is tailored to the subject's condition. OI Type I may be treated in general practice with a particular attention to the DI condition and bisphosphonate intake. Most severe types, such as non-ambulatory type IIIs, may be better served in dedicated treatment centers, due to compromised dentition, severe malocclusions, and general health issues. New pharmacological therapies currently being investigated include teriparatide in adult subjects (*Orwoll et al., 2014*) and denosumab in growing subjects (*Hoyer-Kuhn, Semler & Schoenau, 2014*). Both have seen positive results in recent clinical trials. Mesenchymal stem cell therapy for OI is also under development and preliminary experiments on mouse models show promising results (*Prockop, 2017*).

## ACKNOWLEDGEMENTS

This literature review would not have been possible without the cooperation of the subject's seen under the BBDC 7701 protocol and the help of the McGill Faculty of Dentistry librarian, Mr. Martin Morris. Members of the Brittle Bone Disease Consortium: Michael Bober, Paul Esposito, David R Eyre, Danielle Gomez, Gerald Harris, Tracy Hart, Mahim Jain, Jeffrey Krisher, Sandesh C.S. Nagamani, Eric S. Orwoll, Cathleen L. Raggio, Eric Rush, Peter Smith, Laura Tosi.

### Funding

This work was supported by the Brittle Bone Disorder Consortium (BBDC 7701). The funders had no role in study design, data collection and analysis, decision to publish, or preparation of the manuscript.

## Grant Disclosures

The following grant information was disclosed by the authors:

Brittle Bone Disorder Consortium (BBDC 7701).

## Competing Interests

The authors declare that they have no competing interests.

## Author Contributions

- Maxime Rousseau analyzed the data, contributed reagents/materials/analysis tools, prepared figures and/or tables, authored or reviewed drafts of the paper, approved the final draft.
- Jean-Marc Retrouvey analyzed the data, contributed reagents/materials/analysis tools, prepared figures and/or tables, authored or reviewed drafts of the paper, approved the final draft.

## Data Availability

This article is a literature review and did not generate any data or code.

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
