# Peer review of "Osteogenesis imperfecta: potential therapeutic approaches"

_PeerJ, doi:10.7717/peerj.5464_

## Round 0.1 · original submission · Minor Revisions

Both reviewers provide a number of suggestions that would improve the manuscript and/or clarify certain points. Reviewer 2, in particular, has suggestions on points that would require further discussion or details (interest of complementary examination, such as cone beam computed tomography, before implant therapy, risks of biphosphonate treatment withdrawing, details on the patients described and the implant therapy, potential new pharmacological treatments...)

Please carefully take in account these comments, and revise the manuscript accordingly.

Looking forward to receive your revised version, and with best regards.

·

Basic reporting

This is a well written review article with clear, unambiguous, professional, and technically correct text.
Literature references and contexts are sufficient.
The structure of the article is well format.
This review paper contains a board and extensive on Osteogenesis Imperfecta, bringing useful therapeutic approaches especially for dental practitioners.
I would recommend publication as it unites clinical dentistry with and extensive information in medical research.

As the reviewer, my only criticisms are;
1. Line 46, (sentence)…in the body. Could the authors indicate one or a few references? For instance review articles or about the genetic basis of OI (COL mutations) - useful as PeerJ is a multidisciplinary journal.
2. Line 129, please consider replacing/spelling the abbreviation of “VDO” as I could not find this full name in the manuscript
3. Line 131, I am not quite sure about “Establishing a dental home”. Could you please clarify?
4. Line 138, line-paragraph is broken.
5. Line 191, the word “Osteogenesis Imperfecta” is not in abbreviation. Is it a mistake?
6. Line 218, “Type” - “t” in the lowercase.
7. At references section (line 293-438), there were some journals using the full name and some were not. Please do these consistently regarding the journal format.
8. General comment for Figures: there is no mention of the origin of the pictures/patients shown (from literature? or more likely from the authors clinical archives?) This should be clarified - iconographic sources should be indicated.
9. Figure 2, could you please add more precise explanations such as “Panoramic radiograph of a patient affected by OI type III with missing teeth and DI”?
Text and letter L (bottom right) should not appear on Figure.
10. Table 1, it would be complete if you could insert “Type V: Moderately deforming” (from Rauch F and Glorieux FH, Lancet 2004) in your table because it was missing.

Experimental design

This review article contains both literature articles and their own data with a comprehensive and coverage of the OI.
The organization is logical and simple for general practitioners.

Validity of the findings

The conclusions were appropriately stated and supported by the results.

Additional comments

This review article has collected scientific and clinical relevance of OI. Authors recommended therapeutic approaches respectively on each clinical manifestations of disease. These data indicate that a multidisciplinary team approach is favored in moderate and severe cases as the variability in the degree of severity of OI patients may vary the outcome of treatments.

Overall, this is a well-organized and precise therapeutic guidance for all health professionals especially for dentists. Background evidence is solid and well presented. In addition, the manuscript is clearly written in professional, unambiguous language. If there is a weakness, it is generally in the “Basic reporting” as I have noted above which should be improved upon before Acceptance.

Reviewer 2 ·

Basic reporting

Clear description of the disease (Osteogenesis imperfecta) of clinical interest.
good reason for this review with an adequate introduction of the subject.
Names of the gene (COL1A1 and COL1A2) should be put in italic.
Abstact: line 32: simple extraction can be performed after multi-disciplinary concertation

Experimental design

Exhaustive literature review in the field of osteogenesis imperfect, regarding the clinical, phenotypic and therapeutic aspects. Sources are well cited and recent concerning the clinical studies of biphosphonate treatment effects on jaw bone and implants treatments.

Validity of the findings

Interest of cone beam computed tomography exams could be discussed in the phenotypic characterization of the disease and the pre-surgical imaging, before implants of orthognathic surgery.
Use of zirconia preformed crowns in temporary dentition for esthetic rehabilitation of anterior teeth and also for posterior treatments could be more discussed.
Conclusion coud also open on potential new pharmacological treatments of OI.

Additional comments

line 152: full coverage is preferred when clinical or radiological signes of post-eruptive breakdowns are diagnosed.

line 155: discussion of the clinical difficulties for endodontic therapies in OI patients

line 261: Discuss the risks of biphosphonate treatment withdrawing for OI patients

line 270: add the exhaustive pre-surgical clinical and radiological examinations before implants therapy

line 273: add the number of implants placed in the study and the clinical situations of the patient (number of missing teeth)

---

## Round 0.2 · Minor Revisions

The manuscript has been very carefully revised, taking in account all the reviewers comments. I just found a small inconsistency when checking the manuscript. It appears throughout the main text, the term "patient(s)" has been systematically replaced by "subject(s)". On the other hand, all figure titles and legends refer to "patient(s)". The authors may want to double check if this is intentional or if the same term should be used throughout the manuscript.

I also noticed two small typos in line 280 : ...and five mandibular anterior osseointegrated implants were published. (Prabhu et al. 2007). There was also a report of successful implant therapy in subject
affected by type III OI: there should be no full-stop sign before the Prabhu ref., and the sentence should read:... in a subject affected by....
The authors may want to proof-read again the revised manuscript, especially for the revised sections.

Besides these minor issues, the manuscript is clearly acceptable for publication, and a revised version of the Text file (if necessary) would be sufficient for acceptance.

With best regards,
Pascal Dollé

---

## Round 0.3 · accepted · Accept

Thanks for having corrected these last details. I'm confident this article will be of interest for practitioners and scientists interested in caretaking of osteogenesis imperfecta.

#